# Fabrication and Characterization of Electrospun Chitosan/Polylactic Acid (CH/PLA) Nanofiber Scaffolds for Biomedical Application

**DOI:** 10.3390/jfb14080414

**Published:** 2023-08-05

**Authors:** Yevhen Samokhin, Yuliia Varava, Kateryna Diedkova, Ilya Yanko, Yevheniia Husak, Julia Radwan-Pragłowska, Oksana Pogorielova, Łukasz Janus, Maksym Pogorielov, Viktoriia Korniienko

**Affiliations:** 1Biomedical Research Centre, Sumy State University, R-Korsakova Street, 40007 Sumy, Ukraine; justinsamokhin@gmail.com (Y.S.); yuliia.varava@gmail.com (Y.V.); kateryna.diedkova@lu.lv (K.D.); yanko.ilya.brc@gmail.com (I.Y.); yevheniia.husak@polsl.pl (Y.H.); o.pogorelova@med.sumdu.edu.ua (O.P.); 2Faculty of Chemistry, Silesian University of Technology, 44-100 Gliwice, Poland; 3Institute of Atomic Physics and Spectroscopy, University of Latvia, Jelgavas Iela 3, LV-1004 Riga, Latvia; 4Faculty of Chemical Engineering and Technology, Cracow University of Technology, Warszawska 24 Street, 31-155 Cracow, Poland; julia.radwan-praglowska@pk.edu.pl (J.R.-P.); lukapjanus@gmail.com (Ł.J.)

**Keywords:** chitosan, polylactic acid, electrospinning, nanofibers, biocompatibility, antibacterial biomaterials

## Abstract

The present study demonstrates a strategy for preparing porous composite fibrous materials with superior biocompatibility and antibacterial performance. The findings reveal that the incorporation of PEG into the spinning solutions significantly influences the fiber diameters, morphology, and porous area fraction. The addition of a hydrophilic homopolymer, PEG, into the Ch/PLA spinning solution enhances the hydrophilicity of the resulting materials. The hybrid fibrous materials, comprising Ch modified with PLA and PEG as a co-solvent, along with post-treatment to improve water stability, exhibit a slower rate of degradation (stable, moderate weight loss over 16 weeks) and reduced hydrophobicity (lower contact angle, reaching 21.95 ± 2.17°), rendering them promising for biomedical applications. The antibacterial activity of the membranes is evaluated against Staphylococcus aureus and Escherichia coli, with PEG-containing samples showing a twofold increase in bacterial reduction rate. In vitro cell culture studies demonstrated that PEG-containing materials promote uniform cell attachment, comparable to PEG-free nanofibers. The comprehensive evaluation of these novel materials, which exhibit improved physical, chemical, and biological properties, highlights their potential for biomedical applications in tissue engineering and regenerative medicine.

## 1. Introduction

The electrospinning technique has emerged as a well-established method for producing a wide range of polymers and nanofiber matrices with diameters ranging from 50 to 500 nm [1,2]. This versatile method offers a high surface area to volume ratio fibers, making it applicable in various technological fields, including tissue regeneration [3], biosensors [4], membrane applications [5], and enzyme interactions [6]. Currently, nanofibers are produced by subjecting a polymer solution to a high electric field with carefully controlled process parameters to achieve the desired nanomaterial properties [7].

While numerous research has explored the influence of various electrospinning parameters on the structure of nanofibrous materials, many of them focus solely on the physical characteristics without considering their biological properties. The effects of polymer solution parameters (such as polymer molecular weight and concentration) and process conditions (flow rate, distance between collector and needle tip, applied voltage, temperature, and humidity) on nanofiber size and morphology have been widely acknowledged [8]. However, it is equally crucial to assess the impact of these process parameters on surface morphology, mechanical properties (stress–strain behavior), pore diameter, and filtration properties of nanofiber membranes, as these are important indicators of their quality [9].

Tissue engineering is an area where electrospun biocomposite nanofibers have shown great potential. By mimicking the structure of native tissues, these nanofibers can provide a scaffold for cell growth and differentiation. The ability to tailor the properties of the nanofibers, such as their elasticity and porosity, allows for the development of tissue-specific scaffolds that can support cell attachment, proliferation, and tissue regeneration [10].

A significant advantage of electrospun nanofibers engineering is the ability to fabricate composite polymer fibers with various properties by incorporating different drugs to bring them the required therapeutic properties [11]. Electrospun nanofibers can be loaded with different agents, including nanoparticles, herbal extracts, carbon nanomaterials, and antimicrobial biopolymers, with the aim of advancing antimicrobial properties [12]. In addition, the controlled release of therapeutic agents from electrospun nanofibers is an important area of research. By incorporating drugs or growth factors into the nanofibers, they can be delivered directly to the target site, reducing the need for repeated administration and improving patient compliance [13]. The use of electrospun nanofibers in wound dressings is particularly promising. The large surface area of the nanofibers allows for improved wound healing [14]. The development of Polylactic Acid/Chitosan porous scaffolds was implemented as a potential solution for bone tissue engineering [15].

The limitations related to the electrospun materials comprise poor antimicrobial properties, weak mechanical features, low biodegradability, and biocompatibility can be overcome by blending proteins, polysaccharides, and synthetic polymers depending on requests of soft and hard tissues to regenerate and repair [16,17,18,19]. 

Chitosan (Ch) is a natural polysaccharide widely used as a biomaterial due to its high biocompatibility and the presence of numerous functional groups along its chain. In this study, electrospinning was employed to fabricate nanofibers composed of chitosan/polyethylene oxide (PEO) blends, incorporating polypeptides to enhance cell adhesion [20]. These hydrogen bonds between chitosan and PEO chains enable the electrospinning process, as they help to overcome the entanglements between the chains and facilitate the formation of a stable electrospun jet. This strong interaction between chitosan and PEO also contributes to the improved mechanical properties and stability of the electrospun blends, making them suitable for various applications in tissue engineering and drug delivery [21]. The combination of chitosan and PEO imparts excellent antibacterial and hemostatic properties to the resulting fibers. During the electrospinning process, the inclusion of these water-soluble copolymers enables entanglement with the rigid chitosan chains, resulting in the formation of continuous nonwoven fibers with average diameters ranging from tens to hundreds of nanometers. The presence of nanosized pores within the fibrous mats enhances their permeability, facilitating the exchange of oxygen and nutrients with the surrounding environment. Moreover, the porous structure of the fibrous mat promotes the absorption of wound exudate while reducing the risk of bacterial infections [22].

Poly(lactic acid) (PLA) is a biodegradable aliphatic polyester extensively investigated and utilized in biomedical applications [23]. The composite PLA/chitosan fibers obtained in this study exhibit a porous structure. The results indicate that both the chitosan content and the concentration of the spinning solutions significantly influence the morphology of the porous fibers [24]. The incorporation of chitosan imparts bacteriostatic and bactericidal properties against Escherichia coli and Staphylococcus aureus. Biocompatibility studies have confirmed the non-cytotoxic nature of the materials [25].

PEG (polyethylene glycol) has garnered significant attention in healthcare applications due to its favorable physicochemical properties, low toxicity, and water solubility [26]. PEG-containing PLA-based materials have been employed in various biomedical applications, such as wound-healing patches, local tumor treatment, DNA plasmid delivery, and drug delivery [27,28,29,30]. The solubility of chitosan can be improved by molecular weight control and PEG modification generating ultra-fine fibers by electrospinning of PEGylated chitosan in non-acidic solutions [31]. However, there is a lack of comprehensive studies investigating the effect of PEG incorporation into nanofiber mats on their properties and potential biomedical applications.

Many of the techniques have been explored to enhance the electrospinnability of chitosan/PLA bland. It is important to consider the potential impact of these modifications on the material properties of chitosan, especially in biomedical applications. The tensile strength, biodegradability, and microbial cell affinity of chitosan can be affected by these chemical modifications. Therefore, it is crucial to carefully evaluate and select the appropriate modification techniques to ensure that the desired properties for specific application fields, such as wound dressings or tissue engineering scaffolds, are maintained. Additionally, the modifications should be compatible with the intended use to avoid any potential adverse effects on the interaction between chitosan/PLA nanofibers and bacteria strains that may be encountered in biomedical applications [32]. 

This study aims to investigate the influence of PEG incorporation in Ch/PLA electrospun nanofiber membranes, focusing on their controllable structural and biodegradation parameters, as well as their structural and biological properties. By systematically exploring the role of PEG in these nanofiber membranes, valuable insights can be gained for their potential use in biomedical applications.

## 2. Materials and Methods

### 2.1. Materials

Low-molecular-weight chitosan powder (890,000 Da) was purchased from Glentham Life Sciences (Corsham, UK), CAS 9012-76-4, acetic acid solution (1.0 M), CAS 7732-18-5, was derived from Honeywell (Charlotte, NC, USA). All other reagents—Poly(L-lactide) powder (average Mn 40,000), CAS 26161-42-2, Poly(ethylene oxide) powder (average Mv ~300,000), CAS 25322-68-3, Polyethylene Glycol (MW 1500), CAS 25322-68-3, chloroform (≥99%), CAS 67-66-3, Ethyl alcohol (≥99.8%), CAS 64-17-5, and NaOH (CAS 1310-73-2)—were purchased from Sigma–Aldrich (St. Louis, MO, USA).

### 2.2. Chitosan Solutions Preparation, Electrospinning of Chitosan Nanofibrous Membranes, and Their Alkali Neutralization

10 mL of 99.9% acetic acid (previously dissolved in distilled water to a final concentration of 50% and up to volume 20 mL) and 1.6 g of chitosan powder were mixed, followed by adding 1.6 g of polyethylene oxide (PEO) and stirring [33]. We dissolved 0.2 g of polylactic acid (PLA) in 5 mL of chloroform (after dissolving, the excess chloroform was removed). Afterward, we mix the Ch solution with the dissolved PLA to obtain Solution 1 [34]. The second sample (Solution 2) was received by adding 1.2 g of PEG to the solution prepared identically to Solution 1. Polymer solutions elaboration is represented in Figure 1.

A syringe with a volume of 50 mL (needle inner diameter 0.69 mm) was filled with the polymer solution. The needle-collector distance was 15 cm, the feed rate was 1.5 mL/h, applied voltage was 25 kV. Conditions of electrospinning were as follows: humidity less than 35% and temperature 21–23 °C. Nanofiber membranes were assembled on an electrospinning collector 3 cm in diameter. Chitosan nanofiber membranes were dried at room temperature to remove solvent residues.

As-spun Ch/PLA membranes were treated with 1M sodium hydroxide (NaOH) to decrease their extreme degree of solubility and protect their nanofibrous structure [35]. Both types of membranes (1 − Ch/PLA and 2 − Ch/PLA-PEG) were neutralized with alkali solution 1 M NaOH (70% ethanol/30% aqueous solution) for 12 h, repeatedly washed with distilled water and dried overnight at room temperature. Illustrations of the manufacturing process and functionalization of electrospun nanofibers are shown in Figure 2.

For the bacteriological and cell culture experiment, the samples were decontaminated by soaking in 70% ethanol for 10 min, followed by washing twice in sterile water and placing them under a UV light (at 254 nm) for 30 min on each side.

### 2.3. Scanning Electron Microscopy (SEM)

The structure of the fiber spinning, cell attachment, and bacterial colonization were observed using SEO-SEM Inspect S50-B (SELMI, Sumy, Ukraine). Fiber diameter and ‘local porosity’ were measured using Fiji software (ImageJ 1.51f; Java 1.8.0_102) [36]. ‘Porous area fraction’ was detected with computer binary image analysis. The images were segmented into black (porose) and white (substrate) regions using grey-level thresholding. The ‘porous area fraction’ was defined by the area of the pores divided by the total area of the investigated image region. The ‘porous area fraction’ corresponds directly to a ‘local porosity.’ Frequency histograms of fiber diameter distribution were constructed using Excel software (Office 365 ProPlus).

### 2.4. Surface Characterization-Hydrophobicity 

The surface hydrophilicity of the membrane was monitored by measuring the contact angle (CA) value [37] using a video-based optical contact angle measuring instrument (OCA 15 EC, Data Physics, St. Riverside, CA, USA). The CA value was recorded for ultra-pure water for at least 3 parallel measurements.

### 2.5. Fourier Transform Infrared Spectroscopy (FT-IR)

#### 2.5.1. FT-IR

To show the molecular interactions, chitosan/PLA fibers were characterized by Fourier transform infrared spectroscopy (FTIR). For the experiments, Thermo Nicolet Nexus 470 FT-IR spectrometer equipped with an ATR adapter was used (Thermo Fisher Scientific, Waltham, MA, USA).

#### 2.5.2. Adhesive Properties

To verify the potential of nanofibrous materials in tissue regeneration, their ability to adsorb proteins was verified. For this purpose, samples were immersed in 1 mL of human fibrinogen solution 1 mg/mL and incubated in a CO_2_ incubator for 24 h at 37 °C, 5% CO_2_, and high humidity to imitate in vivo conditions. After 24 h, samples were dried, and FT-IR spectra were collected. To determine the adhesiveness degree based on the absorbance of the genuine fibrinogen solution, solutions leftovers after the incubation study were analyzed by UV-Vis, which was performed using Agilent 8453 Diode array spectrophotometer (Agilent Technologies Deutschland GmbH, 76337 Waldbronn Germany).

### 2.6. Weight Loss (WL)

The weight loss (WL) of the samples (1.0 × 1.0 cm) was determined before and after immersion in PBS. Chitosan membranes were soaked in PBS (pH 7.4) for up to 16 weeks. At each time point, the samples were washed with Milli-Q water and dried overnight at room temperature to remove absorbed water. The following equation [38] was used to calculate the percentage of weight loss (WL) (1):WL (%) = (Ws − Wd)/Ws ∗ 100,(1)
where, Ws (g) is the initial weight of the scaffolds, and Wd (g) is the weight of the samples after drying at room temperature.

### 2.7. Bacteriological Experiment

*E. coli* and *S. aureus* obtained from the National Collection of Microorganisms (DK Zabolotny Institute of Microbiology and Virology, Ukraine) were selected to assess the antibacterial activity of Ch-PEO/PLA fibrous membranes with and without the addition of PEG. The strains were cultured in broth at 37 °C for 24 h. Samples of 0.5 cm^2^ (for bacterial growth-viability assay) and discs of 5 mm in diameter (for disk-diffusion assay).

#### 2.7.1. Bacterial Reduction Rate

Decontaminated samples were placed in a sterile 24-well plastic plate with 2 mL of pre-prepared bacterial broth (10^5^ CFU/mL). Untreated bacteria suspended in nutrient broth were used as controls. After incubation for 2, 4, 6, and 8 h, agar plates were inoculated with aliquots of 10 μL from each well and then incubated at 37 °C overnight to further count the survivors. 

The antibacterial effectiveness was assessed by calculating the antibacterial reduction rate (R) using the following equation [39] (2):R = (C − T)/C × 100,(2)
where C and T are the amounts of surviving bacteria (CFU/mL) in the controls and tested samples, respectively.

#### 2.7.2. Disk Diffusion Method

50 μL of bacterial broth (10^5^ CFU/mL) was spread on a plate of solid MH medium. The disks of the nanofibrous membranes were placed on a plate. The plates were incubated upside down at 37 °C. The diameter of the inhibition zone was measured after 24 h, including the diameter of the disk [40]. Positive control was a disc with cefotaxime. 

#### 2.7.3. Biofilm Formation Study

The samples were sent for examination by SEM to investigate the attachment and proliferation of bacterial strains on the surface of the fibers, followed by the formation of biofilms within the tridimensional structure of the scaffolds after 24 h co-incubation. The sample preparation for SEM was similar to that described previously [41].

### 2.8. In Vitro Cell Culture Study

The mice melanoma B16F10 cell culture was used to evaluate the biocompatibility and biotoxicity of membranes. Materials were sterilized by immersion in 80% ethanol and kept under UV light for 1 h. To remove any remaining ethanol, membranes were then washed three times in phosphate-buffered saline (PBS) for five minutes. The samples were placed in 6-well cell culture plates and immersed in a complete medium overnight. B16F10 mice melanoma cells were cultivated in 25 cm^2^ cell culture flasks under standard conditions of humidified air containing 5% CO_2_ at a temperature of 37 °C. Dulbecco’s modified Eagle medium/nutrient mixture F-12 (DMEM/F-12) with L-glutamine was used, containing 100 units mL^−1^ penicillin, 100 µg mL^−1^ streptomycin, 2.5 µg mL^−1^ amphotericin B, and 10% fetal bovine serum. The cells were seeded on membranes at a density of 10^4^ cells/cm^2^. Wells containing only cells served as a positive control, and those containing only complete medium–negative control. Each sample was conducted in triplicate.

The Alamar blue colorimetric assay was used to assess the total metabolic activity of the cells on the 1st, 3rd, and 5th days. Alamar blue (Sigma-Aldrich, USA) was added to each well in an amount equal to 10% of the medium volume. The plates were incubated for 8 h at 37 °C in the incubator. In a sterile 96-well culture plate, 100 µL of medium from each well was transferred. The absorbance was measured using a microplate reader (Multiskan FC, Thermo Fisher Scientifc, Waltham, MA, USA). plate reader at wavelengths of 570 and 600 nm. The results were imported into Microsoft Excel for further evaluation.

### 2.9. Statistics 

One-way analysis of variance (ANOVA) was performed using Statistica^®^ (v.8 software SPSS Inc., Chicago, IL, USA). Results were expressed as mean ± standard deviation. A *p*-value of less than 0.05 was considered significant. All tests were conducted in triplicate.

## 3. Results

### 3.1. Material Characterization

#### 3.1.1. FT-IR Analysis

Figure 3 presents the results of the FT-IR analysis. All spectra (1–6) exhibit typical chitosan bands. Electrospun membrane prepared from Ch/PLA composite (sample 1; 1) spectrum shows the band coming from hydroxyl groups of chitosan and PEG at 3380 cm^−1^. Two bands are typical for aliphatic moieties typical for all three polymers (-CH_3_ and -CH_2_-) are present at 2882 cm^−1^ and 2866 cm^−1^, respectively. Also, a band of low intensity at 1655 cm^−1^ coming from N-Acetylglucosamine is present. Furthermore, bands characteristic for free amino groups are visible in the region around 1600 cm^−1^ and 1147 cm^−1^. Also, bands that can be assigned to glycosidic bonds and glucopyranose rings at 1099 cm^−1^ and 842 cm^−1^ are visible. No band coming from ester bonds is visible due to the low content of PLA in the membrane. Spectrum 2 shows a membrane prepared from chitosan/PEO and PLA/PEO. Again, bands coming from OH groups are present at 3371 cm^−1^ as well as bands typical for aliphates at 2949 cm^−1^ and 2882 cm^−1^. Moreover, band characteristics for amide bonds around 1650 cm^−1^ as well as for NH_2_ moieties (1597 cm^−1^; 1147 cm^−1^) are visible together with glycosidic bonds between chitosan mers and glucopyranose rings (1061 cm^−1^ and 842 cm^−1^, respectively). Spectra 3 and 4 show membranes treated with an aqueous solution of NaOH. In both cases, it can be noticed that the intensity of bands coming from aliphatic groups (membrane 1 and 2: 2949 cm^−1^; 2879 cm^−1^) has decreased compared to hydroxyl/carboxyl moieties (3319 cm^−1^; 3358 cm^−1^) which suggests degradation of the bonds between polymeric mers, with no visible changes in chitosan structure. Similarly to samples 3 and 4, samples’ 5 and 6 treated with an ethanol solution of sodium hydroxide spectra exhibit increased intensity of the bands coming from -COOH and -OH without a decrease of glycosidic bonds and glucopyranose rings, suggesting that post-processing performed with basic solutions leads to almost negligible changes in the chemical composition of the sample. FT-IR spectra show that it was possible to obtain composite material composed of three different polymers, which combined may be superior to pure chitosan.

#### 3.1.2. Adhesive Properties

Figure 4 reveals the results of the adhesive properties study. Biomaterials dedicated to tissue engineering should exhibit bioactive properties and promote cell proliferation. During the regeneration process, a crucial role plays in various proteins that enable new tissue formation and neovascularization. Therefore, the ability of biomaterials to enhance biomolecule adherence constitutes a very promising feature. As shown in the FT-IR spectrum, all of the newly developed samples exhibit adhesive properties, which confirm the intensity increase of the band’s characteristic for peptide bonds in the region around 1600 cm^−1^, thus proving fibrinogen presence at the membranes’ surface. Adhesive properties can be assigned to the chemical composition of the membranes, especially the presence of amino groups in the chitosan structure, which undergo protonation resulting in the attraction of negatively charged proteins. Moreover, highly developed specific surfaces due to the porosity of the nanofibrous biomaterials additionally positively affect proteins adhesion. Noteworthy, the highest amount of adsorbed fibrinogen can be spotted for samples 5 and 6 (treated with ethanol NaOH solution), which confirms the high intensity of the bands at 1648 cm^−1^ and can be assigned to the high number of hydrophilic moieties. To further investigate adhesive properties, UV-Vis analysis has been carried out. As shown in Figure 5, all samples exhibited the ability to interact with fibrinogen which confirms a decrease in the intensity of the UV-Vis spectra compared to the initial protein aqueous solution. It can be noticed that the highest ability to adsorb fibrinogen exhibits sample 2. Similarly, samples 4 and 6. The worst results were collected in the case of sample 5; however, the adsorption degree exceeds 50%. Obtained results correspond to the data given in Figure 5.

### 3.2. SEM

We successfully obtained nanofibers with a more uniform distribution of fiber diameters and ‘local porosity’ in the chitosan membrane for all samples, as shown in Figure 6. The addition of PLA contributed to the formation of bead-free fibers with a porous structure, indicating good fiber-forming ability [42]. Both membranes exhibited a multilayer structure composed of randomly arranged fiber matrices, as depicted in Figure 7. However, the quantity of obtained fibers varied depending on their size, with fewer fibers observed in samples with smaller fiber sizes. Notably, the presence of PEG resulted in thinner fibers, and the sample with PEG exhibited a higher abundance of nanometer-sized fibers, particularly those below 100 nm in diameter. Additionally, the sample with PEG showed an increased number of pores with sizes below 50 nm^2^.

Our findings demonstrated that the incorporation of PEG into the solution led to a decrease in the average fiber diameter. Moreover, the area fraction of pores decreased compared to the electrospun membrane without PEG. These observations suggest that the presence of PEG improves the electrospinnability of the solution. Furthermore, the increased number of nanoscale fiber diameters can be attributed to the significantly enhanced electrical conductivity of the solution. Consequently, under the same applied voltage and spinning distance, a solution with higher electrical conductivity experiences greater elongation of the jet along its axis, resulting in the formation of fibers with smaller diameters. Overall, these improvements in the electrospinning process contribute to the enhanced quality of the nanofibers [43].

### 3.3. Contact Angle Measurement

The fibers electrospun from the Ch/PLA solution exhibited a wettability value of 54.77 ± 10.38° and from Ch/PLA-PEG solution 51.97 ± 12.11°, as shown in Figure 8. However, it is known that increasing surface hydrophilicity can significantly improve the biocompatibility and functionality of materials [44]. Given the increased surface area of nanoscale fibers, maximizing their surface hydrophilicity becomes particularly important. Therefore, in order to enhance the wettability of the Ch/PLA membrane, it is necessary to adjust the composition and proportion of the blend components. Our test results demonstrate that the presence of PEG improves the hydrophilicity of the membrane, reducing the contact angle to 20.28 ± 3.06° and 22.65 ± 7.40° for as-spun and treated samples, respectively. This improvement can be attributed to the higher hydrophilicity of the PEG block compared to PLA, leading to enhanced overall hydrophilicity [45].

### 3.4. Weight Loss (WL)

When the copolymer comes into contact with water, the molecular chain of PEG breaks down faster than that of PLA. Therefore, the copolymer containing PEG exhibits a better degradation rate [46]. The addition of PEG to the spinning solution did not increase the weight loss of the electrospun membranes during the degradation experiment. Hybrid fibrous materials made from Ch/PLA-PEG co-solvent system demonstrate a slightly lower degradation rate (during the first week of the test) (Figure 9). The process seems to be slightly accelerated in the Ch/PLA samples. However, statistical analysis indicates the degradation profile of PEG-free and PEG-containing fibers are not significantly different. The observed outcome is a result of the secondary interactions that take place between the PLA and the Ch during the dissolution of polymers that can undergo hydrolysis. These interactions produce shorter chains in the acid medium [47].

### 3.5. Antibacterial Effects

#### 3.5.1. Bacterial Reduction Rate

The bacteriological survey results demonstrate that the Ch/PLA-PEG samples exhibit enhanced antibacterial activity against both bacterial species after 4 h and 6 h of incubation, as depicted in Figure 10. Notably, at these time points, *E. coli* shows higher susceptibility to the antibacterial effect of the Ch/PLA-PEG membranes. Specifically, in comparison to the Ch/PLA membranes, the specimens containing PEG exhibit more effective inhibition of bacterial growth after 2 h of co-cultivation with *S. aureus* (*p* < 0.05). However, there is no significant difference between the samples after 24 h of the experiment.

#### 3.5.2. Antimicrobial Assay: Disk Diffusion Susceptibility Test

The results of the antimicrobial testing indicate that the tested samples exhibit varying levels of antimicrobial effects depending on the bacterial strain and the composition of the polymer solution. Specifically, when tested against Gram-negative bacteria (*E. coli*), the fibers containing PEG show a stronger antimicrobial effect compared to the PEG-free samples, as shown in Figure 11. However, the differences in the antibacterial effect between Ch/PLA and Ch/PLA-PEG are minimal when tested against Gram-positive bacteria (*S. aureus*). It is important to note that the observed zone of inhibition may not be extensive if the antibacterial agent, such as chitosan in the sample, is unable to diffuse effectively into the agar medium [48].

#### 3.5.3. Morphology of the Bacteria Biofilm by SEM

In our study, it was observed that S. aureus, a bacterium known for its ability to produce extracellular polymeric substances (EPS), formed a dense biofilm. These EPS molecules have the capability to degrade various polymeric compounds, aiding in surface detachment [49]. The membrane incubated with S. aureus showed the highest concentration of polysaccharides in the biofilm matrix, as depicted in Figure 12. In contrast, when exposed to E. coli, both Ch/PLA and Ch/PLA-PEG membranes did not exhibit any biofilm formation. Only a few bacteria were observed in these cases. The topography of the fiber surfaces plays a significant role in promoting biofilm formation, with factors such as ‘porous area fraction’ and roughness facilitating the migration and adhesion of bacteria to the surface. By incorporating Ch, which possesses antimicrobial properties, into polymer solutions, along with the appropriate combination of polymers, it is possible to effectively reduce bacterial proliferation and minimize the risk of biofilm formation.

### 3.6. In Vitro Cell Culture Study

Total metabolic activity assay results demonstrate that all samples had resazurin reduction levels that are close to the value of the positive control, indicating that membranes tend to be non-toxic to cells (Figure 13). The PEG-containing and PEG-free samples do not significantly differ from one another. It is also important to note that treated samples from both groups exhibit greater resazurin reduction levels than as-spun ones.

## 4. Discussion

Electrospinning is a technique employed to produce biomaterials with desirable structural, mechanical, and biological properties. It allows for the blending of two polymers with distinct properties and structures to enhance the characteristics of the resulting solutions. SEM analysis of the experiment revealed nanofibers of varying sizes, with the composition of the solution influencing their diameter and porous area fraction. Notably, the addition of PEG led to nanofibers with smaller diameters and lower porous area fractions. This demonstrates the potential of combining biocompatible polymers such as PLA and chitosan to create effective scaffolds with biocompatibility for various applications.

To address the hydrophobic nature and low biocompatibility of PLA blends, PEG was utilized. Toncheva et al. revealed that PEG incorporation by physical blending or chemical grafting did not significantly modify the surface wettability of the PLA electrospun mats [50]. Otherwise, further research established proved that blending PEG into a PLA solution is a successful strategy to increase the hydrophilicity of PLA electrospun membranes. Moreover, such an approach improves the water stability of hydrophilic PLA membranes under long-term exposure to aqueous media [51]. Therefore, in our study, Ch/PLA blend showed improved wettability due to added PEG at the pre-electrospinning stage successful strategy to increase the hydrophilicity of Ch/PLA electrospun membranes.

Chitosan, a biodegradable natural polysaccharide derived from marine sources, is an abundant and renewable resource with robust antibacterial properties. The antibacterial mechanism of chitosan is attributed to the presence of positively charged NH2 groups, which hinder bacterial biosynthesis and energy transport through the cell wall, leading to bacterial eradication [52]. In a study by Chengyi Liu et al., PEG, employed as a copolymer to electrospun nanofiber membranes, supported the effectiveness of antibacterial active ingredients. Hydrophilic PEG assisted more antibacterial components in migrating out of the fiber and dissolving in the PBS liquid [53].

Ch/PLA fibrous membranes without PEG exhibited weak bacteriostatic effects, while the inclusion of PEG in the electrospinning solution significantly enhanced the antibacterial activity of Ch/PLA membranes overall. The addition of PEG as a co-solvent has different effects on the antibacterial properties of the samples against *S. aureus* and *E. coli*. Therefore, the difference observed in the antibacterial properties of the samples against Gram-negative and Gram-positive bacteria in our research can be attributed to the specific antibacterial mechanism of action of the compound Ch and its interaction with the distinct cell wall structures of these bacteria [40].

Previous findings suggest that adding hydrophilic polymer PEG to blend for electrospinning holds great potential for controlling and influencing cellular response [54]. PEG-containing PLA fibrous scaffolds supported MSC attachment and proliferation, facilitating MSCs penetration through the interstitial pores. More importantly, the electrospun hybrid PEG/PLA fibrous scaffolds enhanced MSCs’ metabolic activity of the seeded cells, which can be attributed to the enhanced hydrophilic properties of the composite scaffolds [55].

The metabolic activity of hMSCs on scaffolds results indicated that all the scaffolds are cytocompatible. However, a slight increase in the metabolic activity of the seeded cells has been noticed as the concentration of PEG increased, which may have been attributed to the increase in the hydrophilic property of composite scaffolds.

Incorporating polyethylene glycol (PEG) into the blended solution of Ch and PLA serves to control wettability and enhance the biocompatibility of PLA. The as-spun Ch/PLA membranes underwent treatment with 1M sodium hydroxide to reduce their solubility and preserve their nanofibrous structure. It is evident that the addition of PEG maintains the non-toxicity and good biocompatibility of Ch/PLA nanofibers, providing a suitable topographical surface for cell adhesion [56].

## 5. Conclusions

This study provides valuable insights into the role of PEG in enhancing the structural and biological properties of membranes. The incorporation of PLA and PEG as co-solvent in the modification of Ch, along with post-treatment to improve water stability, resulted in hybrid fibrous materials with a moderate degradation rate and reduced hydrophobicity, making them promising for biomedical applications. The Ch/PLA-PEG membranes exhibited improved antibacterial properties over time compared to the PEG-free sample. The optimization of structural parameters, degradation characteristics, and wettability opens up new perspectives for the application of these membranes in biomedical fields and tissue engineering constructs.

## Figures and Tables

**Figure 1 jfb-14-00414-f001:**
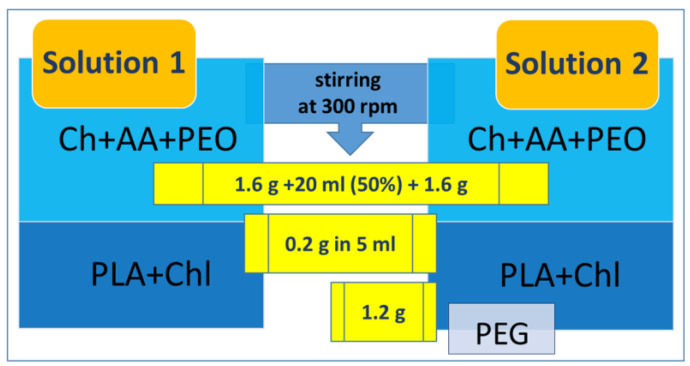
Polymers for electrospinning: Ch/PLA and Ch/PLA-PEG solutions preparation.

**Figure 2 jfb-14-00414-f002:**
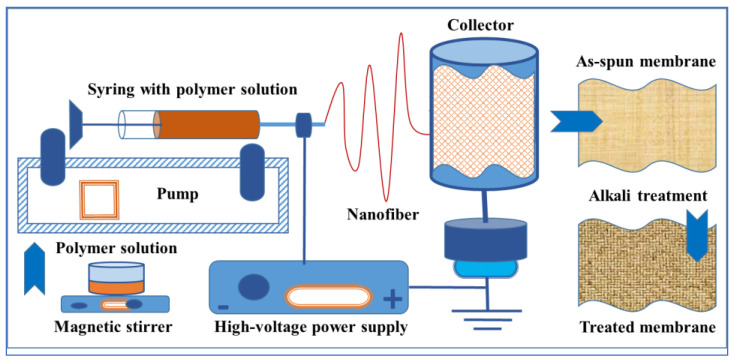
Schematic representation of the manufacturing process and treatment of electrospun nanofibers and fibrous materials by electrospinning.

**Figure 3 jfb-14-00414-f003:**
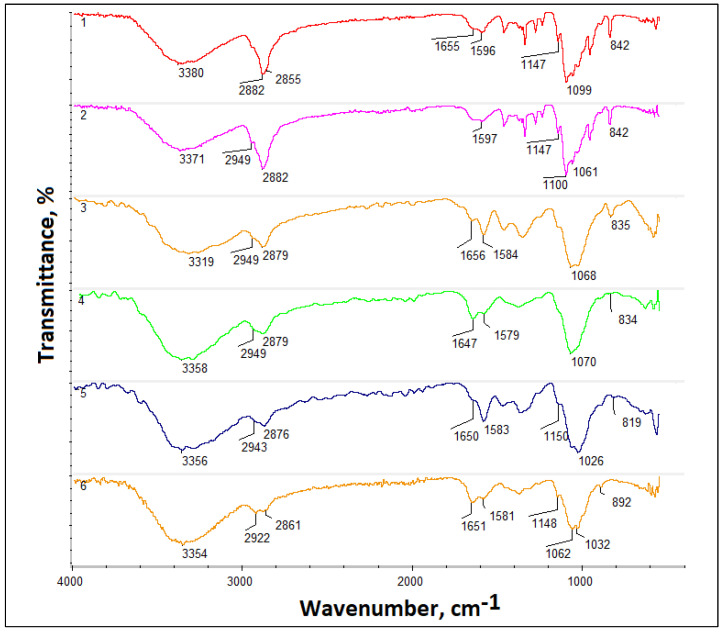
FT-IR analysis (1—as-spun Ch/PLA, 2—as-spun Ch/PLA-PEG, 3–4 treated Ch/PLA, 5–6—treated Ch/PLA-PEG).

**Figure 4 jfb-14-00414-f004:**
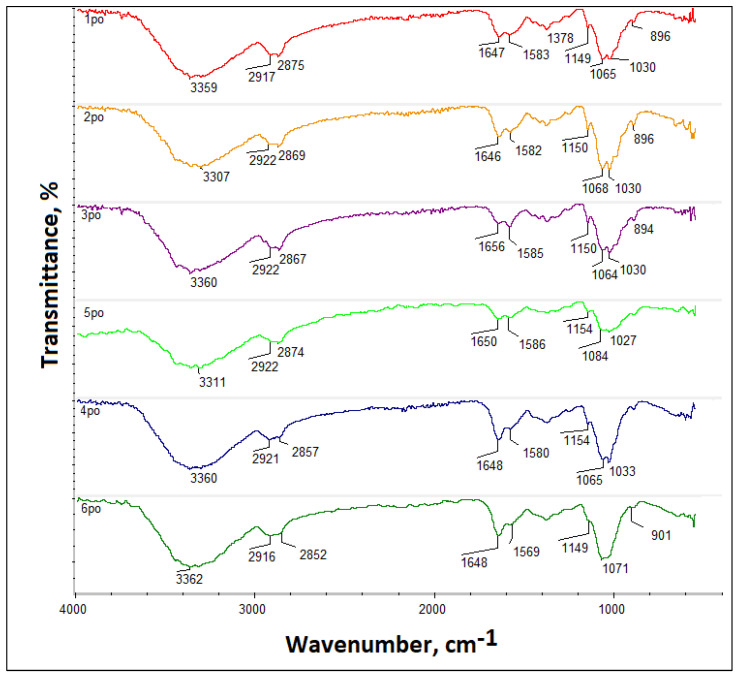
Adhesive properties study (1—as-spun Ch/PLA, 2—as-spun Ch/PLA-PEG, 3–4 treated Ch/PLA, 5–6—treated Ch/PLA-PEG).

**Figure 5 jfb-14-00414-f005:**
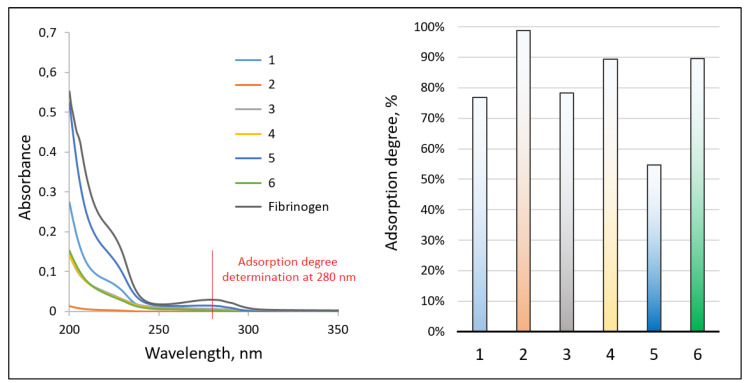
UV-Vis spectra (1—as-spun Ch/PLA, 2—as-spun Ch/PLA-PEG, 3–4 treated Ch/PLA, 5–6 treated Ch/PLA-PEG).

**Figure 6 jfb-14-00414-f006:**
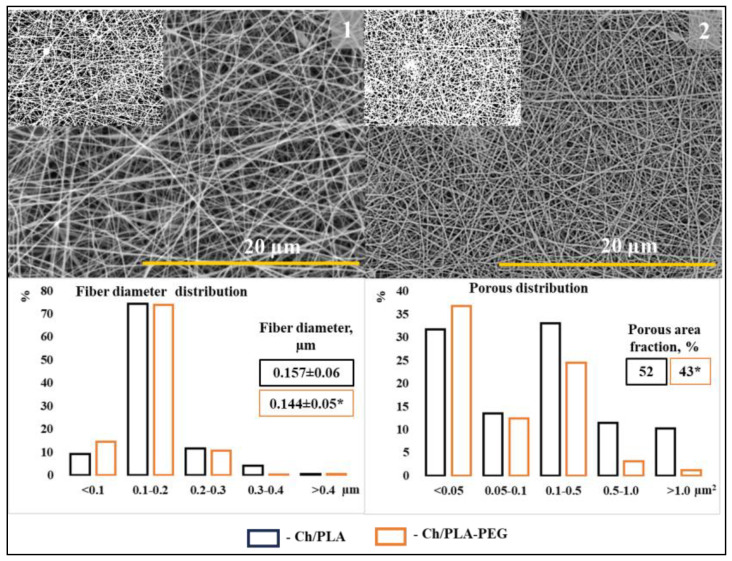
SEM: electrospun membranes (1—Ch/PLA and 2—Ch/PLA-PEG), frequency of “fiber diameter distribution” and “porous area fraction” value of electrospun membranes. The magnification of the main images is ×5.0 K (scale bar = 20 µm). *—denote significant differences between groups at * *p* ≤ 0.05.

**Figure 7 jfb-14-00414-f007:**
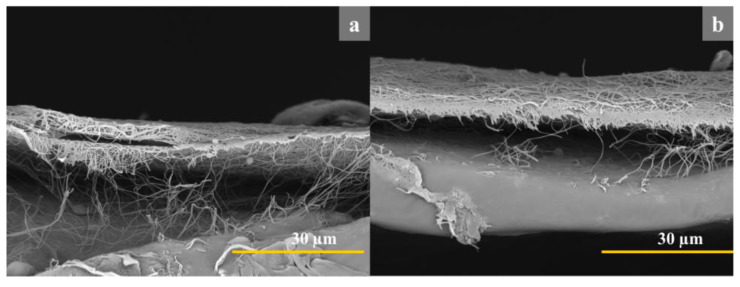
Cross-sectional view of membranes (**a**—Ch/PLA and **b**—Ch/PLA-PEG).

**Figure 8 jfb-14-00414-f008:**
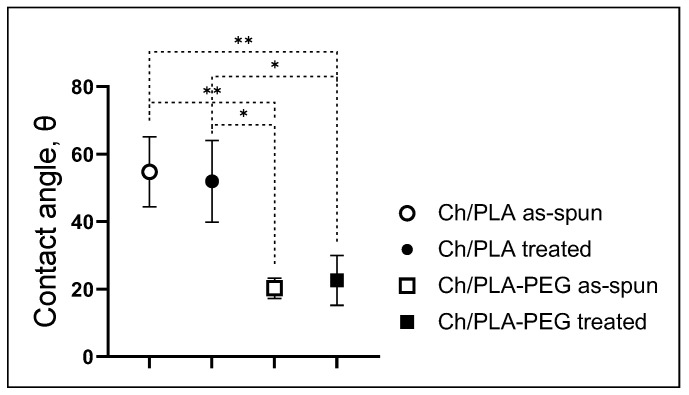
Contact angle/wettability of Ch-PLA membranes − as-spun and treated with NaOH solution. *—denote significant differences between groups at *p* ≤ 0.05; **—*p* ≤ 0.01.

**Figure 9 jfb-14-00414-f009:**
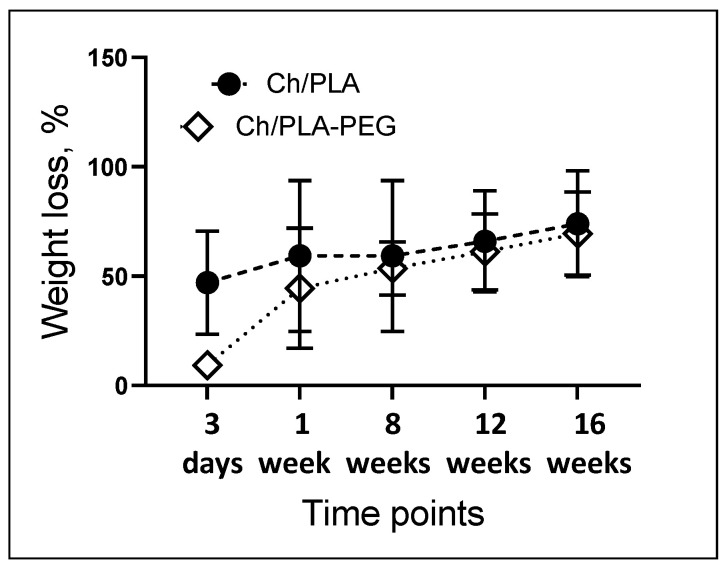
The degradation kinetics of the Ch-PLA membranes after immersion in phosphate-buffered saline (PBS) solution.

**Figure 10 jfb-14-00414-f010:**
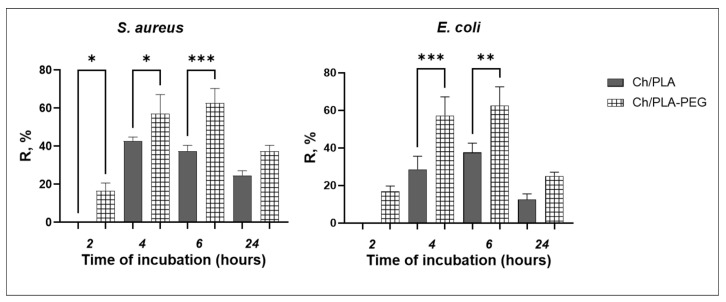
Antibacterial efficiency of Ch/PLA membranes (R—bacterial reduction rate, %). *—denote significant differences between groups at *—*p* ≤ 0.05; **—*p* ≤ 0.01; ***—*p* ≤ 0.001.

**Figure 11 jfb-14-00414-f011:**
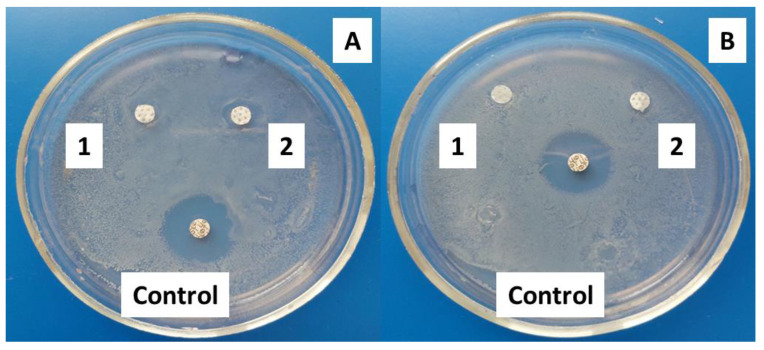
Disk diffusion assay: MH-agar plates after *E. coli* (**A**) and *S. aureus* (**B**) incubation with the electrospun membranes (1—Ch/PLA and 2—Ch/PLA-PEG).

**Figure 12 jfb-14-00414-f012:**
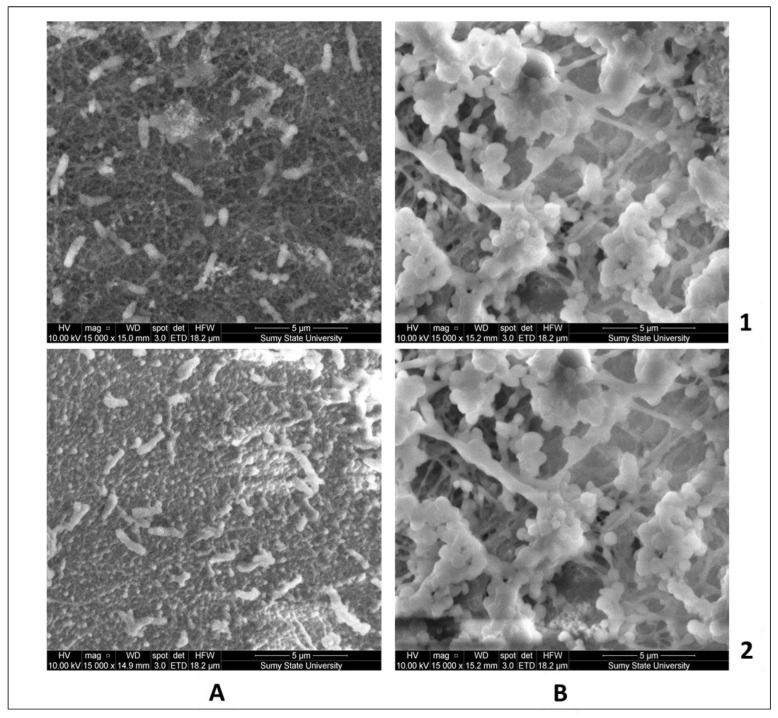
SEM: electrospun membranes (1—Ch/PLA and 2—Ch/PLA-PEG) after 24 h co-cultivation in the bacterial suspension of *E. coli* (**A**) and *S. aureus* (**B**).

**Figure 13 jfb-14-00414-f013:**
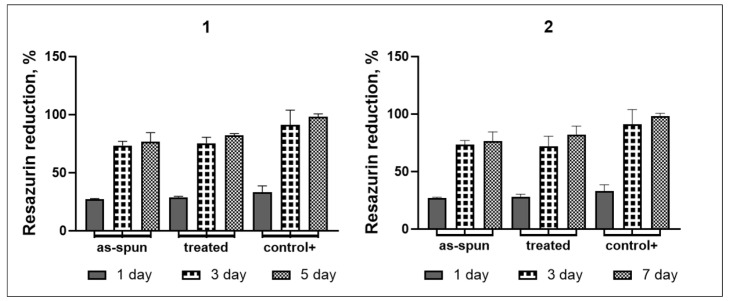
Alamar blue reduction assay data on the proliferation of B16 F10 mice melanoma cells on electrospun membranes (1—Ch/PLA and 2—Ch/PLA-PEG) during the 5-day experiment.

## Data Availability

Data is available on request.

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
