# Peer review of "Fabrication and Characterization of Electrospun Chitosan/Polylactic Acid (CH/PLA) Nanofiber Scaffolds for Biomedical Application"

_jfb, 2023, doi:10.3390/jfb14080414_

Round 1

Reviewer 1 Report

This paper investigates the effect of PEG doping on Ch/PLA electrospun nanofibrous membranes, focusing on their structural and biological properties. I think some explanations should be made in more detail before the paper is published.

1) PCL appears in lines 122, 401 and 403, please check if it is correct and if so add elaboration about PCL.

2) No information about the source of the PLA and ethanol materials is indicated.

3) The purpose of adding PEO and its effect on the system is not stated. Please add a description of PEO in the introduction.

4) Figures 3 and 4 have clarity issues and need to be closed with lines like the other images.

5) There is not enough evidence for the adhesion properties if only Fig. 4 is used.

6) There are no mechanical tests and the mechanical parameters in line 93 are not accurately described.

7) What is the innovation of this article.

Moderate editing of English language required

Author Response

Dear Reviewer,

We thank you and appreciate you taking the time to write a review and bring these issues to our attention, which we believe have served to strengthen the quality of the manuscript.

Sincerely yours,

Author’s team 

  • PCL appears in lines 122, 401 and 403, please check if it is correct and if so add elaboration about PCL.

We thank the Reviewer for the careful revision of our manuscript. We checked and corrected all misprints. We utilized a lot of polymers and in this case it is just typos. 

  • No information about the source of the PLA and ethanol materials is indicated.

We thank the Reviewer for raising this point. We added a material description in chapter 2.1. Materials.

  • The purpose of adding PEO and its effect on the system is not stated. Please add a description of PEO in the introduction.

We thank the Reviewer for raising this point. We add description of PEO importance  in the chapter Introduction.

4) Figures 3 and 4 have clarity issues and need to be closed with lines like the other images.

Dear Reviewer, we tried to do all our best to prepare the figures that fit journal requirements.

5) There is not enough evidence for the adhesion properties if only Fig. 4 is used.

Additional test has been carried out using UV-Vis method to verify % of the protein present in the solution which adhered to the tested materials after 24 h of incubation. New results have been added to the manuscript.

6) There are no mechanical tests and the mechanical parameters in line 93 are not accurately described.

We thank the Reviewer for raising this point. We fixed the text in the chapter Introduction.

7) What is the innovation of this article.

We thank the Reviewer for raising this point. We emphasize the innovation aspects of our research. We add additional description on Introduction section to clarify novelty of our research.

Reviewer 2 Report

This article focuses on the fabrication of composite electrospun nanofibers based on Chitosan and Polylactic acid for biomedical applications. Research on nanofibers fabrication based on biodegradable polymers has gained significant attention in the scientific community, but it also presents several challenges. The article presents a comprehensive study covering various aspects, including fiber fabrication, physicochemical characterization, antibacterial properties, and in-vitro experiments. However, several points in the article require clarification:

1. In the Abstract section, the authors mention "PEG as a co-solvent" (line 23, page 1). Is this supposed to be "PEG and PEO as a co-solvent"? The FI-IR analysis results show the peak of PEG, so can PEG be considered a solvent or does it form a bond with the polymer? It would be helpful if the authors could explain this.

2. Line 47, page 2 states that "flow rate and distance between collector and needle tip" are not solution parameters.

3. In the Ch/PLA nanofiber preparation, the total concentration of Chitosan and Polyethylene oxide was 32 w/v %. With such a concentration it is quite high and difficult to dissolve Chitosan with a molecular weight of 890,000 Da.  How did the author dissolve Chitosan?

4. As described in the fiber preparation section, sample 1 consisted of Ch and PLA only; sample 2 has only Ch; PLA; PEG without PEO. Can the author elaborate on this and provide citations?

5. Did the authors optimize the electrospinning parameters for fabrication nanofiber? If not, could the authors provide references or literature on selecting polymer concentration and technology parameters of electrospinning for Chitosan nanofiber fabrication?

6. When drying the nanofiber membrane at room temperature, is all the solvent removed? How did the authors control this process?

7. The results regarding contact angle show a significant difference of 20.38° and 2.17° (lines 306, 307, page 9). Could the authors explain these results? The standard deviation results in Figure 7 seem inconsistent with the description on page 9.

8. Could the authors provide more details on the method of measuring porosity using ImageJ software? Assessing porosity through images seems questionable for a three-dimensional nanofiber membrane.

9. In lines 401, 403, 404 on page 13, PCL is mentioned. Is this an error?

10. The units throughout the manuscript need to be corrected. For example, in line 153 on page 4, lines 169 and 204 on page 5, and throughout page 6, units such as cm2, °C, and cm-1 need to be used consistently and appropriately.

Author Response

Dear Reviewer,

We thank you and appreciate you taking the time to write a review and bring these issues to our attention, which we believe have served to strengthen the quality of the manuscript.

Sincerely yours,

Author’s team

  1. In the Abstract section, the authors mention "PEG as a co-solvent" (line 23, page 1). Is this supposed to be "PEG and PEO as a co-solvent"? The FI-IR analysis results show the peak of PEG, so can PEG be considered a solvent or does it form a bond with the polymer? It would be helpful if the authors could explain this.

Dear Reviewer, we fixed statements about co-solvents: “PLA and PEG as a co-solvent”. Polymers acted as an additional component which created homogenous composite material with chitosan/PLA. It does not create chemical bonds with chitosan, however it may interact to create hydrogen ones with OH and NH2 groups of the biopolymer.

  1. Line 47, page 2 states that "flow rate and distance between collector and needle tip" are not solution parameters.

Dear Reviewer, thank you for this important comment. We corrected points concerning parameters.

  1. In the Ch/PLA nanofiber preparation, the total concentration of Chitosan and Polyethylene oxide was 32 w/v %. With such a concentration it is quite high and difficult to dissolve Chitosan with a molecular weight of 890,000 Da.  How did the author dissolve Chitosan?

We would like to thank the Reviewer for the suggestion. Chitosan powder was first dissolved in acetic acid (10 ml of 99,6% acid was diluted to 50%), and then polyethylene oxide was added and mixed. We indicated all the required information about solution preparation in Figure 1.

  1. As described in the fiber preparation section, sample 1 consisted of Ch and PLA only; sample 2 has only Ch; PLA; PEG without PEO. Can the author elaborate on this and provide citations?

We thank the Reviewer for raising this point. We revised the description of solution preparation and clarify it and corrected Figure 2 as well.

  1. Did the authors optimize the electrospinning parameters for fabrication nanofiber? If not, could the authors provide references or literature on selecting polymer concentration and technology parameters of electrospinning for Chitosan nanofiber fabrication?

We would like to thank the Reviewer for her/his suggestion. We optimized the electrospinning parameters for the fabrication of nanofibers based on our previous experience and data from other researchers (added references are marked in yellow color).

  1. When drying the nanofiber membrane at room temperature, is all the solvent removed? How did the authors control this process?

 We thank the Reviewer for raising this point. To overcome this issue of supposed toxic effect of residual solvents we proved the biocompatibility of nanofibers in cell culture experiments. At this moment we did not control the remnant of solvent residual after electrospinning, but cell culture data demonstrate membrane safety and biocompatibility.

  1. The results regarding contact angle show a significant difference of 20.38° and 2.17° (lines 306, 307, page 9). Could the authors explain these results? The standard deviation results in Figure 7 seem inconsistent with the description on page 9.

We thank the Reviewer for the careful revision of our manuscript. We checked and fixed the contact angel data in the text to provide a clear and correct description of Figure 7.

  1. Could the authors provide more details on the method of measuring porosity using ImageJ software? Assessing porosity through images seems questionable for a three-dimensional nanofiber membrane.

We greatly appreciate your thoughtful comments that helped improve the manuscript. We clarified the term porosity as local porosity (porous area fraction) and explained its measurement.

  1. In lines 401, 403, 404 on page 13, PCL is mentioned. Is this an error?

We thank the Reviewer for the careful revision of our manuscript. PCL was mentioned by mistake. We checked and corrected all misprints.

  1. The units throughout the manuscript need to be corrected. For example, in line 153 on page 4, lines 169 and 204 on page 5, and throughout page 6, units such as cm2, °C, and cm-1need to be used consistently and appropriately.

We thank the Reviewer for the careful revision of our manuscript. We checked and corrected units through the text of the manuscript.

Reviewer 3 Report

This study aims to investigate the influence of PEG incorporation in Ch/PLA electrospun nanofiber membranes, focusing on their controllable mechanical and biodegradation parameters, as well as their structural and biological properties. By systematically exploring the role of PEG in these nanofiber membranes, valuable insights can be gained for their potential use in biomedical applications.

1. Discuss the uses of this scaffold in different fields of biomedicine.
2. There are many studies in which
Chitosan/Polylactic Acid (CH/PLA) NanoFibe has been synthesized and its effects have been investigated. What is the difference in its structure or effects with previous studies?
3. What drugs can be loaded on this nanofiber?
4. The manuscript should be updated with similar new references (2022-2023).
5. No references are provided for any of the methods.
6. Write the possible disadvantages and limitations of using this fiber in biomedicine in the discussion section.
7. The comparison of this research with other studies in the discussion has not been done and the discussion is very short and incomplete.
8. Manuscript English writing must be checked completely for grammar.
9. In line 184, the number of cells should be corrected (it seems that in number 105, the number 5 is a superscript).
10. What does R % mean in figure 9?
11. The following articles will enrich the manuscript:
https://doi.org/10.1080/10837450.2019.1656238
  https://doi.org/10.3390/polym14091661
https://doi.org/10.3390/ddc2010010

Manuscript English writing must be checked completely for grammar.

Author Response

Dear Reviewer,

We thank you and appreciate you taking the time to write a review and bring these issues to our attention, which we believe have served to strengthen the quality of the manuscript.

Sincerely yours,

Author’s team

1. Discuss the uses of this scaffold in different fields of biomedicine.
2. There are many studies in which Chitosan/Polylactic Acid (CH/PLA) NanoFibe has been synthesized and its effects have been investigated. What is the difference in its structure or effects with previous studies?
3. What drugs can be loaded on this nanofiber?

We thank the Reviewer for raising this points (questions 1-3). We revised introduction and add  appropriate references to the list. Added references are marked in yellow color.

  1. The manuscript should be updated with similar new references (2022-2023).
  2. No references are provided for any of the methods.
  3. Write the possible disadvantages and limitations of using this fiber in biomedicine in the discussion section.
    7. The comparison of this research with other studies in the discussion has not been done and the discussion is very short and incomplete.

We thank the Reviewer for the careful revision of our manuscript. We revised the manuscript and added appropriate references (for question 4-7). Added references are marked in yellow color.

  1. Manuscript English writing must be checked completely for grammar.

We carefully checked grammar throughout the manuscript.

  1. In line 184, the number of cells should be corrected (it seems that in number 105, the number 5 is a superscript).

We thank the Reviewer for the careful revision of our manuscript. We checked and corrected units through the text of manuscript.

  1. What does R % mean in figure 9?

We thank the Reviewer for raising this point. We indicateted meening of R - bacterial reduction rate in figure 9.

  1. The following articles will enrich the manuscript:
    https://doi.org/10.1080/10837450.2019.1656238
    https://doi.org/10.3390/polym14091661
    https://doi.org/10.3390/ddc2010010

We thank the Reviewer for the careful revision of our manuscript. We revised the manuscript and added appropriate references. Added references are marked in yellow color.

Round 2

Reviewer 1 Report

I agree to accept this manuscript.

I agree to accept this manuscript.

Reviewer 2 Report

The author has solved all the problems that I have posed. Articles can be published in current form.

Reviewer 3 Report

Corrections have been made and the manuscript is acceptable.